# NEAT1 and Paraspeckles in Cancer Development and Chemoresistance

**DOI:** 10.3390/ncrna6040043

**Published:** 2020-10-30

**Authors:** Gabriel Pisani, Byron Baron

**Affiliations:** Centre for Molecular Medicine and Biobanking, University of Malta, 2080 Msida, Malta; gabrielpisani2@gmail.com

**Keywords:** NEAT1, paraspeckle, RNA-binding proteins, 3′ processing variants

## Abstract

Non-coding RNA were previously thought to be biologically useless molecules arising from simple transcriptional noise. These are now known to be an integral part of cellular biology and pathology. The wide range of RNA molecules have a diverse range of structures, functions, and mechanisms of action. However, structural long non-coding RNAs (lncRNAs) are a particular class of ncRNA that are proving themselves more and more important in cellular biology, as the exact structures that such RNAs form and stabilise become more understood. Nuclear Enriched Abundant Transcript 1 (NEAT1) is a specific structural RNA emerging as a critical component in the progress and development of cancer. NEAT1 forms part of multiple biological pathways, acting through a diverse group of mechanisms. The most important of these is the formation of the paraspeckle, through which it can influence the stability of a tumour to develop resistance to drugs. This review will thus cover the range of effects by which NEAT1 interacts with cancer progression in order to describe the various roles of NEAT1 in chemoresistance, as well as to identify drug targets that protein research alone could not provide.

## 1. Introduction

Whilst the paraspeckle, which is the compartment within the nucleus that Nuclear Enriched Abundant Transcript 1 (NEAT1) forms, was discovered in HeLa cells in 2002 [1], NEAT1 itself was only discovered in 2007 [2]. It is located in the multiple endocrine neoplasia locus in the human chromosomal region 11qa [3]. Since then, the roles of NEAT1 and the paraspeckle have expanded greatly to also include the interplay between these factors and cancer chemoresistance. In order to understand the importance of NEAT1 in cancer pathology, the structure and biological roles of NEAT1 and paraspeckles will be addressed.

### 1.1. NEAT1 and Paraspeckle Structure

The *NEAT1* gene produces two transcripts NEAT1_1 and NEAT1_2, which differ in their 3′ untranslated region (UTR) processing. NEAT1_1 is the result of polyadenylation and canonical RNA processing, forming a 3.7 kb transcript, and NEAT1_2 undergoes a non-canonical form of RNA processing, forming a triple helix structure, which is then cleaved by RNase P forming a 23 kb transcript [4]. Distinguishing between these isoforms is crucial since they have different roles and functions as will be explained in depth later. NEAT1_1, despite its tendency to form microspeckles [5], is not essential for paraspeckle formation [4]. On the other hand, NEAT1_2 is not only necessary for paraspeckle formation, but is also the limiting factor in paraspeckle formation, and, thus, the tendency of a nucleus to form paraspeckles is dependent on the concentration of NEAT1_2 [6]. NEAT1_2 is a structural long non-coding RNA (lncRNA) capable of interacting with RNA-binding proteins (RBPs) to form the paraspeckle. It has three domains: an A domain that aids in stability [7], a B domain involved in isoform switching [8], and a C domain that interacts with RBPs. The C domain is used to bind to a heterodimer, consisting of polypyrimidine tract-binding protein-associated splicing factor (P54^nrb^, which is the human homologue of the mouse protein NONO) and splicing factor, proline- and glutamine-rich (SPFQ), in order to initiate paraspeckle formation. These subsequently recruit further dimers through the coiled coil (CC) domain. These dimers then likely aid NEAT1_2 to recruit other proteins with the prion-like domain. This domain acts through weak electrostatic forces to form an aggregate of many more paraspeckle proteins, which form a region denser than the surroundings. This achieves liquid–liquid phase separation to form a sub-structure, which, even though is not separated from the surrounding nucleoplasm with membranes, allows for a certain degree of separation. This process is further aided by the chromatin remodelling complex switch/sucrose non-fermentable (SWI/SNF) and RNA–RNA interactions [8]. There are 37 RBPs that associate with the paraspeckle, seven of which are required for paraspeckle function including P54^nrb^, SPFQ, SWI, RNA Binding Motif Protein 14 (RBM14), heterogeneous nuclear ribonucleoprotein K (HNRNPK), fused in sarcoma (FUS), Deleted in Azoospermia Associated Protein 1 (DAZAP1), and heterogeneous nuclear ribonucleoprotein H3 (HNRNPH3) [4].

### 1.2. Mechanistic Roles of NEAT1

NEAT1_2 functions by regulating the formation of the paraspeckle. The paraspeckle can regulate genes by three main mechanisms. Firstly, the formation of the paraspeckle itself alters the distribution of the paraspeckle proteins, which have diverse sets of nuclear roles. SPFQ is of particular relevance to chemoresistance since it functions as a transcription factor, activating several apoptotic genes, such as B-cell Chronic Lymphocytic Leukemia/Lymphoma 2 (BCL2) binding component 3 (BBC3) and BCL2 associated X protein (BAX), the expression of which is prevented when SPFQ is retained in the paraspeckle [9]. In fact, this pathway is likely to play a role in repressing chronic myeloid leukaemia (CML). In CML, the *BCR–ABL* fusion gene activates master regulator of cell cycle entry and proliferative metabolism (c-Myc). This in turn represses *NEAT1,* so that no paraspeckles form, and SPFQ can increase the ability of cells to undergo apoptosis [9]. This means that disruption of this pathway aids in CML development, such that the tumour cells develop paraspeckles to sequester SPFQ and decrease apoptosis. As a result, aberrant paraspeckle formation in CML results in tumours becoming more resistant to apoptosis induced by chemotherapies or radiotherapy. This pathway is summarised in Figure 1 below. Similarly, the other paraspeckle proteins also have diverse functions. For instance, P54^nrb^ and SWI are complex nuclear proteins involved in many processes [10,11]. However, in the case of these proteins, the ways their sequestration affects cancer development is as yet unknown.

The second mechanism is through RNA retention. Paraspeckle proteins, such as SPFQ or P54^nrb^, can bind to RNA transcripts which undergo extensive RNA editing, specifically adenine-inosine (A-I) editing. The paraspeckle typically retains such RNA so that it cannot leave to undergo translation, such as with cationic amino acid transporter 2 (CAT-2) [12]. This mechanism can also slowly accumulate RNA transcripts in ways that do not significantly alter gene expression but then release all of the transcripts at once to result in a dramatic increase in messenger RNA (mRNA), as is the case for the F11 receptor (F11R) transcript [13]. The extent to which this mechanism influences the gene expression of the cell as a whole, and whether this process regulates a majority of transcripts or just a select amount has yet to be determined, especially with reference to how it affects chemoresistance. The third mechanism is by interacting with miRNA. The paraspeckle has been shown to interact with microRNAs (miRNAs), since certain miRNA processors, such as DiGeorge syndrome chromosomal region 8 (DGCR8), tend to localise in the paraspeckle [14]. However, the exact interaction that occurs is still unknown. Furthermore, the transcripts of *NEAT1* can act in ways that are independent of the paraspeckle itself. Firstly, NEAT1 can act as a competitive endogenous RNA (ceRNA), also referred to as a miRNA sponge. This means that the product of *NEAT1* (probably NEAT1_1) binds to miRNAs to prevent them from binding and repressing mRNAs [15,16,17]. The product of *NEAT1* can also bind and activate a variety of proteins by acting as an RNA scaffold for nuclear proteins. An example of this is the recruitment of enhancer of zeste homolog 2 (EZH2) in the polycomb repressive complex 2 (PRC2) [18] as well as DNA methyltransferase 1 (DNMT1) [19], an enzyme that methylates DNA, a modification resulting in gene suppression [20]. However, the specific isoform of *NEAT1* responsible for such functions remains unknown.

## 2. NEAT1 in Cancer and Chemoresistance

With these previously mentioned pathways in mind, the involvement of paraspeckles in the complex array of factors that constitute cancer can be described. Although the full extent of the interactions have not yet been identified, enough research has been carried out in the past few years to begin to address the ways in which paraspeckles are involved in chemoresistance. The current knowledge presents a complex picture in which *NEAT1* and even the paraspeckles interact with cancer in different ways. In some cases, these promote (and other times inhibit) cancer growth, resulting in *NEAT1* increasing and decreasing chemoresistance in different biological contexts. That being said, certain prognostic and even therapeutic targets have been identified.

### 2.1. The Role of Paraspeckles in Tumour Progression and Chemoresistance

In the previous section, the established participation of paraspeckles in cancer was demonstrated using the CML context, where *BCR–ABL*, the main cause of a vast majority of CMLs [21], tends to activate c-Myc, which then represses NEAT1, which in turn prevents paraspeckle formation. This, as explained previously, allows SPFQ to resume its role as a transcription factor for apoptosis-related genes, promoting apoptosis in cancer cells [9]. Therefore, paraspeckles can be viewed as promoting cancer due to their role in repressing apoptosis. Thus, there is a possibility that re-establishing this pathway could have therapeutic benefits. Paraspeckles have also been shown to increase resistance to apoptosis due to changes in miRNA patterns, although the exact interactions have not been identified. Increased levels of miRNA will typically lead to more double-stranded RNA (dsRNA), which can cause apoptosis. Too little miRNA will cause interferon stimulated genes (ISGs) to be expressed, which also results in apoptosis. The paraspeckle has been shown to prevent this and stabilise miRNA concentrations [22,23,24]. The way this affects cancer has not been shown; however, it could possibly inhibit miRNA-induced apoptosis, allowing tumour cells to mutate in ways that alter the presence of miRNA without killing the cell. This would result in a cell that is more able to withstand random mutations, such as those undergone constantly throughout the cancer evolution process. This also means that paraspeckles effectively increase the ability of cancer cells to withstand apoptotic chemotherapies, increasing their chemoresistance potential. Additionally, paraspeckles have been shown to be induced under hypoxic conditions [25]. Hypoxia is a well-established pathway that tumours utilise to increase their invasive, proliferative, and chemoresistant properties [26]. Hypoxia inducible factor 2 (HIF-2) has been shown to activate *NEAT1* expression and drive the formation of paraspeckles. This means that paraspeckles could be one of the downstream mediators of the hypoxic pathway [25]. This leads to the identification of NEAT1_2 as a possible prognostic marker due to its ability to indicate the extent of the hypoxic response within cells. There is also the possibility to serve as a therapeutic target in order to prevent cancer cells from activating the hypoxic response, which tends to result in increased chemoresistance [26]. However, it is important to note that this study did not distinguish between either isoform of *NEAT1* leaving the possibility that the isoform NEAT1_1 is the target of HIF-2 induced gene expression of *NEAT1*. This means NEAT1_1 is the downstream factor in the hypoxic pathway and, therefore, it is NEAT1_1 that could be increasing the chemoresistance of a tumour, whilst the increased paraspeckle formation is just a side product [25]. This will be discussed further in Section 2.2. Furthermore, paraspeckles have been shown to be an important component to the DNA damage response in oncogenic cells [27]. NEAT1_2 suppression was shown to bring about a disruption of Ataxia telangiectasia, mutated (ATM) signalling, resulting in a weakened DNA damage response (DDR) and, thus, in enhanced replication stress [27]. This means that NEAT1_2 expression and the resulting paraspeckle formation in tumour cells results in a greater resistance to chemotherapies that cause replication stress. Furthermore, it is important to note that in this study NEAT1_2 was specifically suppressed instead of simply preventing *NEAT1* expression in general. This is important because by downregulating the isoform that actually forms the paraspeckle it eliminates the possibility of NEAT1_1 producing effects not related to the paraspeckle. Despite this however, *NEAT1* and by extension, the paraspeckle has also been shown to suppress tumour formation. This is due to its interaction with p53, a protein heavily involved in tumour prevention. This protein is involved in multiple pathways related to the prevention of cancer such as those resulting in apoptosis and senescence [28]. Moreover, p53 also tends to suppress tumour progression without halting the cell cycle or killing the cells in ways that have not yet been fully described [29]. Recently, *NEAT1* has been shown to be a highly significant downstream regulator of this p53 tumour suppressor pathway [30,31,32], which results in tumour suppression and decreased chemoresistance. In fact, a p53 binding motif was found in the promotor of *NEAT1* and activation of p53 resulted in the expression of *NEAT1* [30]. This study also demonstrated that attenuation of *NEAT1* resulted in a decreased ability for p53 to supress tumours, proving that *NEAT1* is partly responsible for the tumour suppressor effects of p53 [30]. Furthermore, in pancreatic cancer, the loss of *NEAT1* increases the rate of acinar-to ductile metaplasia (ADM), a process of dedifferentiation, which is an early stage of pancreatic cancer formation [30]. This is due to *NEAT1* aiding in the expression of several pancreatic differentiation factors, such as Basic Helix-Loop-Helix Family Member A15 (Bhlha15) and SRY-Box Transcription Factor 9 (SOX9) [30], as well as tumour suppressor genes that tend to inhibit cancer progression, such as Plexin A4 (Plxna4), which is involved in the prevention of vascular endothelial growth factor (VEGF), a key protein in angiogenesis and growth [33]. Therefore, the induction of Plxna4 expression by *NEAT1* results in decreased proliferative abilities and decreased angiogenic properties of tumours, through the inhibition of VEGF [34]. The role of *NEAT1* in expressing differentiation factors is especially important when considering cancer stem cells (CSCs). This subpopulation present in tumours can proliferate quicker, are more invasive and are more resistant to drugs [35]. The latter is due to numerous factors including the presence of the ATP binding cassette (ABC) family of membrane proteins designed to protect stem cells. Such proteins are crucial to prevent the accumulation of foreign substances in stem cells, which need to conserve the integrity of their DNA sequences in order to pass on to daughter cells DNA that have not been mutated. However, such proteins in CSCs promote chemoresistance by preventing drugs from entering the cell [36]. This could potentially link paraspeckles with the prevention of CSC formation, which could possibly be a very useful therapeutic target to prevent CSC development, and thus significantly decreases chemoresistance. This role of the *NEAT1* transcripts in promoting differentiation is interesting since in smooth muscle, paraspeckles have the opposite effect, tending instead to promote dedifferentiation through sequestering WD Repeat Domain 5 (WDR5) within the paraspeckle, resulting in a decrease in the expression of smooth muscle genes [37]. This further highlights the important notion that many interactions are cell-specific and that the paraspeckle may have a different or even an opposite effect in different cellular contexts. Considering the previously mentioned chemoresistance inhibiting properties, it is no surprise that NEAT1_2, which is the long 3′ processing variant that forms the paraspeckle, is generally considered a tumour suppressor, where lower levels indicate a poorer prognosis such as in colorectal cancer and can, thus, serve as a potential prognostic marker for the disease since higher. In fact, NEAT1_2 levels have been linked with better prognosis, due to it lowering the chemoresistance of tumours [38].

### 2.2. Distinguishing the Roles of NEAT1 3′ Processing Variants in Cancer Progression and Their Impact on Chemoresistance

Despite the fact that NEAT1_2 variant has been considered as a tumour suppressor, the *NEAT1* gene is generally considered an oncogene that results in increased chemoresistance (Table 1). In fact, expression of *NEAT1* (which isoform was not specified) increased tumour growth and chemoresistance in a variety of cancers, such as breast cancer [15,39] and non-small cell lung cancer (NSCLC) [15,40]. This can be further demonstrated by the wide variety of oncogenic factors that tend to activate the transcription of *NEAT1* (both isoforms seem to be activated), such as the previously mentioned HIF-2 as well as octamer-binding transcription factor 4 (OCT4) [40]. *NEAT1* also forms part of the epidermal growth factor (EGF) response [41]. Furthermore, it is repressed by tumour suppressors, such as the breast cancer gene 1 (BRCA1) [39]. In fact the transcripts levels of *NEAT1* are generally elevated in tumours leading to both isoforms of *NEAT1* being characterised as a potential biomarker for cancer [38]. *NEAT1* levels (without distinguishing between isoforms) are also increased in the blood of cancer patients and this has been linked to increased *NEAT1* levels in immune cells especially neutrophils [38]. Further research into the role of *NEAT1* isoforms in the immune system will be required and could potentially reveal further insights into how the immune system combats cancer. In order to reconcile the increased chemoresistance due to an increase in the total *NEAT1* transcript levels but the decreased chemoresistance brought about by increased NEAT1_2 levels, it has to be postulated that NEAT1_1 is responsible for the oncogenic processes mentioned (those caused when *NEAT1* expression increases due to OCT4, EGF, and H1F-2, whilst being repressed by BRAC1), as shown in Figure 2. In this scenario, the roles of *NEAT1* that are independent of the paraspeckle, explained in Section 1.2, are all assumed to be carried out by NEAT1_1 and these are, in turn, responsible for the pathways associated with increased chemoresistance of NEAT1. This would explain why NEAT1_1 is considered oncogenic while NEAT1_2 is generally considered a tumour suppressor. Additionally, the hypoxic response and the p53 response are found in all cell types and have opposite effects, making it unlikely that both use the paraspeckle as a part of the pathway, adding further evidence that NEAT1_1 is, thus, likely to be the variant forming part of the hypoxic pathway. Additionally, when both NEAT1_1 and NEAT1_2 levels are elevated, the prognosis of a patient is poor suggesting the possibility that NEAT1_1 masks the tumour supressing effect of NEAT1_2 [38]. This data supports the interpretation that paraspeckles, when formed under hypoxic conditions, do not actually contribute to the pathway, but rather it is NEAT1_1 through miRNA sponging and interaction with epigenetic proteins, such as PCR2, which results in a hypoxic response. The paraspeckles then act as part of a tumour suppressor pathway downstream of p53 to decrease cancer cell growth, which is unfortunately ineffective due to the high concentration of NEAT1_1, leading to an overall increase in chemoresistance [38]. However, a recent paper found no correlation between cancer growth and the NEAT1_1 isoform [42]. There are various possible explanations for this, for example the fact that the cells where treated with DNA-damaging agents, such as doxorubicin, which would induce replicative stress. Such stress could shift the tumour priorities from growth and spreading to surviving the replicative stress, which could mask the effects of NEAT1_1 on tumour cells growth. Additionally, the effects of NEAT1_1 could be specific to different cancers [42]. All of this means that more experiments evaluating the role of NEAT1_1 in chemoresistance are needed to further understand the complex interplay between NEAT1_1 and chemoresistance.

Since the product of *NEAT1* (not specified in the research) is involved in methylation, and the fact that this methylation is oncogenic, it is possible that the NEAT1_1 isoform is involved in the methylation of the gene for miR-129-5p on the chromosomal locus 7q32.1 [43], preventing the repression of Wingless-related integration site family member 4 (WNT4), which is an important factor in maintaining stem cell properties [39]. This pathway is used by cancer cells to not only enable rapid proliferation, but also develop and maintain the presence of CSCs [39], resulting in higher chemoresistance. This suggests that the *NEAT1* isoforms might have different effects on CSC development with NEAT1_1 aiding in CSC development and NEAT1_2 generally opposing CSC development, such as in the pancreas [37]. NEAT1_1 could therefore also be responsible for gene methylation that induces p53 repression. Therefore, it is possible that *NEAT1* exists in a negative feedback loop where p53 induces *NEAT1* for paraspeckle formation through NEAT1_2, which aids tumour suppression, whilst NEAT1_1 is also expressed and reduces p53 expression to form the negative feedback loop [19]. Further roles of *NEAT1* in cancer that are likely due to the NEAT1_1 isoform are summarised in Table 1.

### 2.3. Alternative 3′ End Processing of NEAT1 as a Therapeutic Target for Chemoresistance

Considering the alternative effects that both isoforms may have on chemoresistance, altering the 3′ end processing of *NEAT1* could be a potentially useful therapeutic target, as it would not only eliminate the presence of the tumour progressing NEAT1_1, but would also replace it with the tumour repressing NEAT1_2. This would therefore serve as a target to decrease chemoresistance through two means at once. The 3′ alternative processing of *NEAT1* is yet to be fully understood, however it involves cleavage and polyadenylation specificity factor subunit 6 (CPSF6) and nudix hydrolase 21 (Nudt21) [4]. These form a heterodimer that binds to a cluster of UGUA repeats slightly upstream of the polyadenylation start site (PAS) and then recruits more proteins for the canonical polyadenylation complex to the transcript to form NEAT1_1. Alternatively, HNRNPK, another paraspeckle protein binds to pyrimidine rich regions such a CU-rich region between the UGUA repeats and the PAS. HNRNPK then tends to compete with CPSF6 for binding to Nudt21. The more HNRNPK-Nudt21 binding that occurs, the greater the amount of NEAT1_2 formed relative to NEAT1_1 [4,8] (Figure 3). Other factors, such as the B domain on the *NEAT1* transcript, and the other paraspeckle proteins, such as P54^nrb^/NONO are known to play a role in the process but the interplay between these factors and the 3′ end processing is not yet understood [4,8]. Interestingly, it was also shown that when Nudt21 levels are decreased, NEAT1_2 levels are markedly increased and NEAT1_1 levels are decreased. Therefore, Nudt21 or other elements of this alternative 3′ end-processing pathway could possibly be used as therapeutic targets targeting the differential 3′ processing between the two isoforms. The value in such a therapy is exemplified by the interaction between *NEAT1* and p53. As previously explained, NEAT1_2 mediates downstream tumour suppression of p53 [30] whilst a product of *NEAT1* (likely NEAT1_1) inhibits p53 through methylation of the TP53 gene [19]. Therefore, it can be hypothesised that if the 3′ processing variants were altered to increase NEAT1_2 at the expense of NEAT1_1, p53 repression would probably be reduced and the downstream effectors of p53 would be activated to a greater extent. This could overall result in a significantly stronger p53 response, decreasing chemoresistance greatly. Further experiments are required to determine how beneficial such a therapy could possibly be, which cancer subtypes would possibly benefit, and which chemotherapies could possibly work synergistically with such treatment. This could possibly prove to be useful for the clinical treatment of various cancers, as it could possibly be used alongside other therapies for a stronger response.

## 3. Conclusions

Considering the available data, *NEAT1* clearly illustrates the importance of lncRNA in cancer and chemoresistance. By being the key structural component to the paraspeckles, as well as through other means, *NEAT1* interacts with gene regulatory pathways to bring about changes in gene expression that increase or decrease the ability of tumours to withstand chemotherapy and form CSCs. Thus, the transcript levels of *NEAT1* could be a potential biomarker of diagnostic and prognostic value. Despite the fact that a number of the functions of *NEAT1* in tumours are unknown, research is beginning to show that the two 3′ processing variants, NEAT1_1 and NEAT1_2, generally have opposing effects in tumours. Even though more studies, which take into account the specific isoform involved in the process investigated are needed to confirm this, the current data is beginning to reveal that by altering the ratio of such 3′ processing variants an effective therapy to decrease the chemoresistance of a tumour may possibly emerge.

## Figures and Tables

**Figure 1 ncrna-06-00043-f001:**
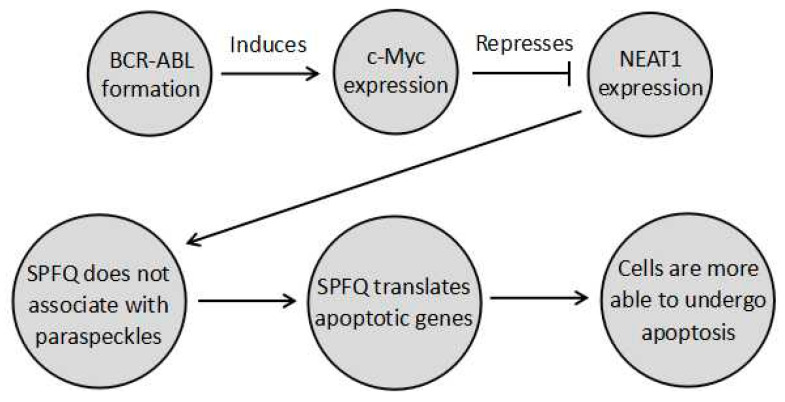
A pathway undergone by chronic myeloid leukaemia (CML) cells where *BCR–ABL* prevents paraspeckle formation through cell cycle entry and proliferative metabolism (c-Myc) induction, so that splicing factor, proline- and glutamine-rich (SPFQ) is no longer sequestered in the paraspeckle. It can instead transcribe apoptosis-related genes to increase the ability of cells to undergo apoptosis. Modulating this pathway in combination with chemotherapy could possibly provide a means to increase the sensitivity of CML to apoptotic chemotherapy, by decreasing chemoresistance.

**Figure 2 ncrna-06-00043-f002:**
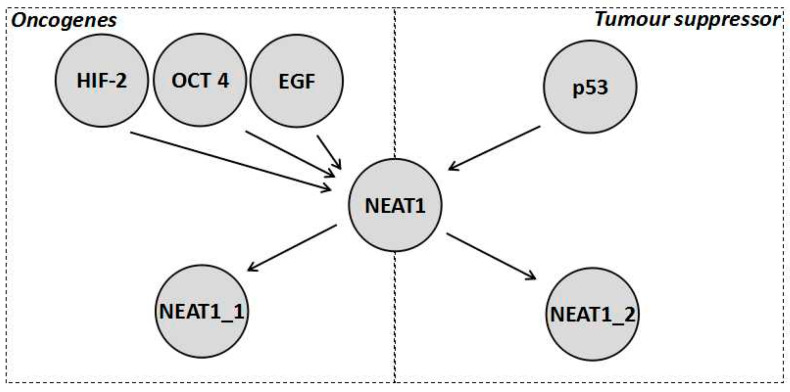
Diagrammatic representation of a reconciliation of the current data about the role of *NEAT1* in chemoresistance where *NEAT1* is activated by numerous factors both tumour promoting and tumour inhibiting. As shown, NEAT1_1 mitigates the oncogenic functions, which is why it is a marker for worse prognosis in colorectal cancer, whilst NEAT1_2 mitigates the tumour suppressor functions. However, it should be noted how both are activated by oncogenic stimulation, which is why paraspeckles tend to form on stimulation of HIF-2, yet its tumour suppressor effect is not enough to balance the oncogenic effect of NEAT1_1.

**Figure 3 ncrna-06-00043-f003:**
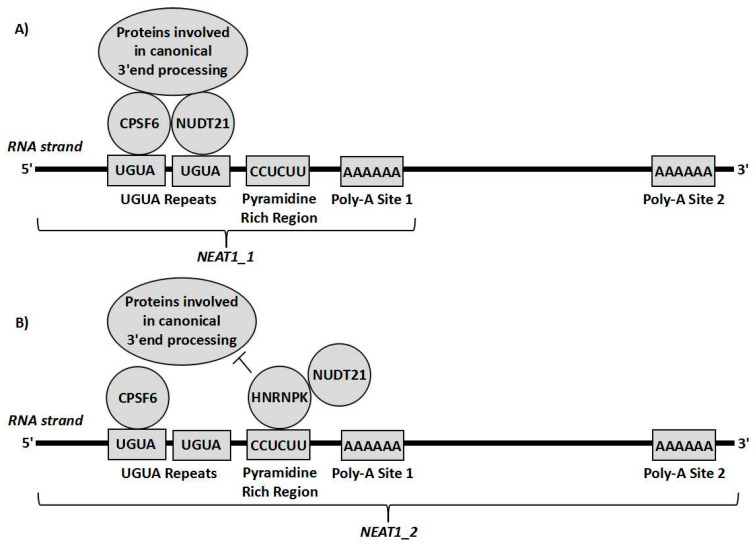
Representation of cleavage and polyadenylation specificity factor subunit 6 (CPSF6) and nudix hydrolase 21 (Nudt21) forming a heterodimer on the CUG repeats upstream of the polyadenylation site and recruiting proteins involved in polyadenylation so that the transcript undergoes canonical RNA processing forming NEAT1_1 (**A**). Alternatively, HNRNPK can bind to the pyrimidine rich site close to the UGUA repeats and interact with Nudt21, sequestering it from CPSF6, so that the proteins needed for canonical RNA processing cannot act, leading to the transcript being cleaved by RNase P resulting in the NEAT1_2 isoform (**B**).

**Table 1 ncrna-06-00043-t001:** A table showing the interactions *NEAT1* undergoes independent of paraspeckles, and, thus, likely performed by the isoform NEAT1_1, which promote more rapid tumour growth and, thus, a greater resistance to chemotherapy. This is mostly mediated through miRNA sponging but should also be noted that since each miRNA tends to regulate several proteins, only the main protein/proteins involved in tumorigenesis are mentioned, despite the fact that changes in the expression levels of the other unmentioned proteins may also affect tumour development.

Cancer Type	Role of *NEAT1* in Promoting Tumour Proliferation
Breast Cancer	*NEAT1* decreases expression of the miRNA’s miR-129-5p, miR-101, miR-211, and miR-448 through miRNA sponging of the latter 3 and methylation of the gene for miR-129-5p. This increases the expression of WNT4 [39], EZH2 [44], AT-hook 2 (HMGA2) [45] and Zinc finger E-box-binding homeobox 1 ZEB1 [46] all of which result in more aggressive breast cancer and make it more able to undergo epithelial to mesenchymal (EMT) transition, resulting in higher chemoresistance. Additionally, *NEAT1* aids in the formation of a complex between forkhead/winged helix transcription factor N3 (FOXN3) and paired amphipathic helix protein 3 (SIN3A), which represses EMT by supressing genes like trans-acting T cell-specific transcription factor 3 (GATA3) [47]. *NEAT1* has also been identified to play roles in angiogenesis and glycolysis through interactions with transforming growth factor beta 1 (TGF-β1) and Lactate dehydrogenase A (LDHA) [48], respectively.
Cervical Cancer	*NEAT1* decreases miR-193b-30 and miR101 levels, which allows for increased cyclin D1 [49] and AP-1 Transcription Factor Subunit (FOS) [50], expression, respectively, promoting unregulated cellular division resulting in a more aggressive cancer.
Clear Cell Renal Cancer	*NEAT1* inhibits miR-34a-5p from repressing hepatocyte growth factor receptor (HGFR) [51], resulting in an increase in tumour progression.
Gastric Cancer	*NEAT1* binds to miR-506-3p decreasing miR-506-3p levels to increase STAT3 levels towards increased tumour development [52].
Glioblastoma	*NEAT1* binds to miR-107 miR-7e-5p thereby preventing them from repressing CDK6 [53] and neuroblastoma RAS (NRAS) [54], resulting in increased cancer growth as the rate of cell division increases.
Hepatocellular Carcinoma	*NEAT1* decreases the abundance of miR-129-5p targeting valosin-containing protein (VCP) and an inhibitor of I_k_B, which in turn is an inhibitor of nuclear factor kappa-light-chain-enhancer of activated B cells (NF_K_B). This results in increased levels of NF_K_B and VCP, which in turn increase the rate of tumour growth [55]. *NEAT1* also sponges miR-613, miR-485, and miR-139-5p, which result in increased expression of doublecortin-like kinase 1 (DCLK1) [56], signal transducer and activator of transcription 3 (STAT3) [57] and TGF-β1 [58]. *NEAT1* also interacts with miR124-3p to increase lipolysis in order to promote cancer progression through its interaction with Adipose triglyceride lipase (ATGL) [59].
Oral Squamous Cell Cancer	*NEAT1* supresses miR-129-5p and miR-365-3p by acting as a ceRNA, leading to the expression of C-terminal-binding protein 2 (CTBP2) [60] and Regulator of G Protein Signalling 20 (RGS20) [61], respectively, aiding in tumour development and promoting chemoresistance.
Osteosarcoma	*NEAT1* prevents miR-34c-5p and miR-194-5p from repressing BCL2 [62] and Cyclin D1 (CCND1) [63], respectively, leading to increased cancer growth.
Ovarian Cancer	*NEAT1* inhibits miR-34a5p, miR-194 and miR-382-3p, thus, resulting in the increased expression of B-cell lymphoma-2 (BCL-2) [64], zinc finger E-box-binding homeobox 1 (ZEB1) [65] and Rho-associated coiled-coil containing protein kinase 1 (ROCK1) [66], respectively, which are oncogenic and increase chemoresistant properties.
Prostate Cancer	*NEAT1* interacts with steroid receptor coactivator-3 (SCR-3) in order to increase the transcription of insulin like growth factor 1 (IGFR1), leading to increased tumour development [67]. *NEAT1* also increases expression of cell division cycle 5-like protein (CDC5L), which acts as a transcription factor to promote the expression of AGRN, which interacts with TGF-β1 to decrease cell cycle arrest by decreasing DNA damage and promotes cancer growth [68].
Nasopharyngeal Carcinoma	*NEAT1* inhibits miRNA miRlet-7a-5p and miR-124 resulting in an increase in the RAS-MAPK signalling pathway [69] and NF-_k_B [70] respectively, increasing tumour migration and proliferation.
Non-Small Cell Lung Cancer	*NEAT1* acts as an miRNA sponge decreasing the levels of miR-181a-5p, miR-377-3p and miR-98-5p, which results in increased expression of high-mobility group box 2 (HMGB2) [71], E2F Transcription Factor 3 (E2F3) [72] and mitogen-activated protein kinase 6 (MAPK6) [73] respectively, which all result in increased survival and proliferation in tumours. Additionally, *NEAT1* activates WNT4 but whether this occurs through miR-129-5p repression or through some other mechanism is unknown [74]. *NEAT1* also binds to DNMT1 to induce the DNA methylation of the p53 gene and the cyclic GMP-AMP synthase (cGAS) gene bringing about an increase in tumour proliferation [19].

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
