# Peer review of "NEAT1 and Paraspeckles in Cancer Development and Chemoresistance"

_ncrna, 2020, doi:10.3390/ncrna6040043_

Round 1

Reviewer 1 Report

This is a really interesting review and a commendable effort to pull together some the very disparate (and confusing!) literature on how NEAT1 is involved in cancer. While the review poses some interesting ideas, I think it could use some work to improve clarity and support their case, in particular for the naiive reader. My suggestions are below.

Lack of clarity
It is often difficult to tell which way the signaling is going. Use of terms like “regulates” should be paired with positive or negative or some sort of direction. For instance, on lines 159-163, the text appears to be suggesting that NEAT1 aids in the expression of PlexinA4, which down-regulates VEGF, which enhances the angiogenic properties of tumors. This doesn’t make any sense.

I’d also like to see more discussion of what was measured in each paper. As the authors are discussing NEAT1_1 versus NEAT1_2 effects in cancer, it is vital to assess and discuss what was actually measured in the papers the reference.

Similarly, it is absolutely essential to be very clear when the author is talking about evidence for the involvement of the paraspeckle (likely only NEAT1_2), NEAT1 in general or specifically NEAT1_1 or NEAT1_2. An example of confusing text is lines 150-165. The paragraph starts out discussing interactions of the paraspeckle with p53 but ends by saying “* the extent to which the paraspeckle itself is involved in these interactions with p53 as opposed to NEAT1_1 is not yet known” If that is the case, the paragraph should start out talking about NEAT1 in general, not the paraspeckle.

Another way to improve things might be to explicitly discuss in each interaction in section 2.1 which isoforms could be involved. For example, when talking about oncogenic conditions increasing paraspeckles (eg. Hypoxia?), the authors could note that oncogenic stimulation turns on the NEAT1 locus, producing both NEAT1_1 and NEAT1_2 (as outlined in Figure 2) and paraspeckles.

Line 176, is it accurate (and clearer) to say “this role of NEAT1 in promoting differentiation” rather than “suppressing dedifferentiation”?

Contradictory statements

The review contains some puzzling apparent contradictory statements that should be resolved.
For example:

* Is Paragraph 182-186 (“all the above mentioned interactions considered NEAT1_2”) only referring to the tumor suppressor interactions or to all the interactions in the 2.1 section of the paper? If the latter, it directly contradicts the line of argument laid out in paragraph 203-220, as if the evidence in Section 2.1 is for paraspeckles and NEAT1_2, then the NEAT1_2 interactions include both pro- and anti-tumorigenic interactions.

* In lines 210-213, NEAT1_1 (i..e the non-paraspeckle forming NEAT1 variant) is suggested to be responsible for he hypoxic pathway interaction, but on lines 141, the authors speak of paraspeckles being induced under hypoxic conditions. These ideas appear to contradict each other.

Overstating the case
The authors should be careful not to overstate their case. In particular in Section 2.2, it would wise of the authors to be clearer in separating supposition from fact.

For example:

* Lines 206-210. It is possible, given the presumptive separate localization and structural roles of NEAT1_1 and NEAT1_2, not highly likely.

* On lines 214-215, the say “Additionally, when both NEAT1_1 and NEAT1_2 levels are elevated, the prognosis of a patient is poor suggesting that NEAT1_1 masks the tumour suppressing effect of NEAT1_2 [31]. “, It would be more prudent to say “Additionally, when both NEAT1_1 and NEAT1_2 levels are elevated, the prognosis of a patient is poor suggesting the possibility that NEAT1_1 masks the tumour suppressing effect of NEAT1_2 [31]. “,

Line 234 : “ This means that the NEAT1 isoforms have different effects on CSC development with NEAT1_1 aiding in CSC development and NEAT1_2 generally opposing CSC development such as in the pancreas “
Would be more accurate to say “ This suggests that the NEAT1 isoforms might have different effects on CSC development with NEAT1_1 aiding in CSC development and NEAT1_2 generally opposing CSC development such as in the pancreas “

* Line 250: “ Considering the alternative effects that both isoforms have on chemoresistance, “ should more accurately say “considering the alternative effects that both isoforms may have on chemoresistance,

* In Table 1, evidence that NEAT1_1 is responsible for the effects rather than NEAT1_2 should be presented

Missing literature

I could find no mention of such recent, highly relevant literature as the Dec 2019 Adriaens et al. paper in RNA showing NEAT1_1 to be dispensable for normal tissue homeostasis and cancer cell growth. The authors should explore more of the literature where NEAT1_1 and NEAT1_2 were explicitly assayed separately.

Author Response

We thank Reviewer 1 for the positive comments on our manuscript. We have made an effort to improve the clarity of the text overall based on the provided comments.

Lack of clarity
The terms “regulates” is only used in instances where a general statement is being made and the signalling can go both ways. 

Lines 159-163 have been reworded to clarify that NEAT1 aids in the expression of PlexinA4, which down-regulates VEGF, which reduces the angiogenic properties of tumours.

Unfortunately most papers do not measure specifically the levels of NEAT1_1 versus NEAT1_2.

We have clarified as much as possible when talking about evidence regarding paraspeckles or specific isoforms of NEAT1, including lines 150-165. 

In Section 2.1 all the papers included have the common theme that they claim that paraspeckles cause the effects observed even though they don’t distinguish between isoforms. This has been specifically mentioned now to clarify that a lot of paraspeckle research does not distinguish between isoforms or just claim that NEAT1 expression was increased.

The process of “suppressing dedifferentiation” in line 176 is different from the proposed “promoting differentiation” as the former is the inhibition of going from a differentiated to an undifferentiated state while the latter is going from an undifferentiated or early differentiated state to a fully differentiated state.

Contradictory statements

* In paragraph 182-186 we changed the text to “Considering the previously mentioned chemoresistance inhibiting properties” to specify.

* In both lines 210-213 and line 141, the pathway being referred to is the induction of paraspeckles under hypoxic conditions. The researchers that conducted this study concluded that paraspeckles are part of the hypoxia pathway. There is reason to believe that this is just a biproduct and that NEAT1_1 is the real reason why hypoxia induces expression of the NEAT1 gene.

Overstating the case
We have worked to separate the data from the extrapolations, not only in Section 2.2, but throughout.

* We have reworded lines 203-210 more in line with the suggestion of Reviewer 1 to use “possible” instead of “highly likely”.

* We have added “the possibility” to lines 214-215 as suggested. 

* We have changed line 234 to “This suggests that the NEAT1 isoforms might have different effects on CSC development with NEAT1_1 aiding in CSC development and NEAT1_2 generally opposing CSC development such as in the pancreas “ as suggested.

* Added the word “may” to line 250 as suggested

Missing literature

We have searched for additional references related to chemoresistance where NEAT1_1 and NEAT1_2 were explicitly assayed separately but could not find any. If Reviewer 1 has any papers in mind, please let us know.

Reviewer 2 Report

The review manuscript ncrna-821586 by Pisani and Baron represents very well summarized role of NEAT1 and paraspeckles in chemoresistance.  

Author Response

We thank reviewer for going through our manuscript 

Reviewer 3 Report

The review entitled "NEAT1 and paraspeckles in chemoresistance - the importance of structural RNA in cancer development and chemoresistance" is a slightly modified version of a review recently published by the same authors doi: 10.1016/j.ncrna.2019.11.002. The review is therefore not very original.

Major concerns:

-The graphics are totally unappealing

-English is poor and the text contains several typos

-Science is sketchy and superficial. A few example below, but the list is way longer: only 1 out 3 important papers showing a connection with p53 pathway have been cited, and is not even the first one describing this link.

In lane 219 the authors say that p53 interacts with paraspeckles. But NEAT1 does NOT interact with p53 nor p53 localises to paraspeckles.

Also the switch from NEAT1_1 and NEAT1_2 is called here alterantive splicing, but it isn't actually. The switch is caused by alternative 3'end processing.

Author Response

We thank Reviewer 3 for the comments. We have tried to address all comments as best we could.

Reviewer 3 stated that our manuscript “is a slightly modified version of a review recently published by the same authors doi: 10.1016/j.ncrna.2019.11.002”. This review is actually a build up of our previous work. The first review covers “paraspeckles for apoptotic regulation”. The word “chemoresistance” is not mentioned once. We hope Reviewer 3 will reconsider this opinion. 

We fully understand that the graphics are not attractive visually. They were created with the intention to simplify and summarise the text, to improve the reader’s understanding. 

We have corrected any typos found in the text and have tried to improve the overall clarity of the text also in line with the request of Reviewer 1. 

We have included additional references showing a connection between paraspeckles and the p53 pathway.

We have replaced the confusing phrase In line 219 with “The paraspeckles then act as part of a tumour suppressor pathway downstream of p53 to decrease cancer cell growth, which is unfortunately ineffective due to the high concentration of NEAT1_1”.

Any mentions of alterantive splicing in section 2.3 have been changed to “alternative 3' end processing”.

Reviewer 4 Report

This is an interesting review on NEAT1 and paraspeckles in cancer development and chemoresistance. 

However, in its current state, this manuscript is a challenge for the general reader. The organization is challenging, currently NEAT1_2, NEAT1_1, then NEAT1 (not always clear which transcript). Paragraphs are highly variable in size and there are very few topic sentences to guide each paragraph. Many sentences are far too long, notably in the abstract. Table 1 could be in alphabetical order. Figs 1 and 2 don't add more than the text (Fig. 3 was helpful though). There are numerous typographical errors.

Scientifically, the authors have picked 77 references from approx. 330 references on NEAT1 and cancer in pubmed, so this seems reasonably representative. Some references e.g. lines 194-199 are peripherally related to cancer and could be trimmed. Paraspeckles are not defined.

Author Response

We thank Reviewer 4 for the comments. We have tried to address all comments as best we could.

We have corrected any typos found in the text and have tried to improve the overall clarity of the text also in line with the request of Reviewer 1 and Reviewer 3.

We have reduced the length of some sentences including in the Abstract by breaking them up.

We have clarified as much as possible when talking about evidence regarding paraspeckles or specific isoforms of NEAT1. Unfortunately most papers do not measure specifically the levels of NEAT1_1 versus NEAT1_2.

We have reorganised Table 1 in alphabetical order. 

Figs 1 and 2 were not intended to add more to the text but to summarise the process graphically. 

We did not understand what is being requested by “Paraspeckles are not defined”. In Line 28 we open with “paraspeckle, which is the structure that NEAT1 forms” and then Section 1.1 covers specifically “NEAT1 and paraspeckle structure”. If the Reviewer gives us a bit more detail to this request, we will be happy to make the appropriate changes.

Round 2

Reviewer 1 Report

I greatly appreciate the revisions by the authors, the text is clearer and I was better able to understand the evidence behind the ideas presented.

Author Response

Dear Reviewer 1, 

Thank you for going through our manuscript again

Reviewer 4 Report

The authors have addressed most of the requested comments.

i) Referencing is the major issue. These jump from #18 in line 106, to # 34 in line 109, to #70 in line 109. There is no intervening table or figure that would explain this jump.

ii) The opening sentence "Whilst the paraspeckle, which is the structure NETA1 forms" doesn't really say what a paraspeckle is. It would help the general reader to define a paraspeckle as a subnuclear body or other relevant text.

iii) Minor: line 30: 11qa or 11q?  line 45: long non-coding RNA rather than RNAs. line 58 RBPs rather than RPBs

Author Response

Dear Reviewer,

The references 68-74 were added after the completion of the manuscript which is why they are not in order. We have now reordered all the references.

We have added a short define of a paraspeckle in line 28.

Line 30 is correct. We have corrected the typos in lines 45 and 58